# Breeding for Prolificacy, Total Carotenoids and Resistance to Downy Mildew in Small-Ear Waxy Corn by Modified Mass Selection

**Supaporn Sukto [1,2]**, **Khomsorn Lomthaisong [3]**, **Jirawat Sanitchon [1,4]**, **Sompong Chankaew [1,4]**, **Shanerin Falab [5]**, **Thomas Lübberstedt [6]**, **Kamol Lertrat [4]** and **Khundej Suriharn [1,4,*]**

1   Department of Agronomy, Faculty of Agriculture, Khon Kaen University, Khon Kaen 40002, Thailand; supaporn.suk@gmail.com (S.S.); jirawat@kku.ac.th (J.S.); somchan@kku.ac.th (S.C.)
2   Department of Agriculture (DOA), Uthai Thani Agricultural Research and Development Center, Uthai Thani 61110, Thailand
3   Department of Biochemistry, Faculty of Science, Khon Kaen University, Khon Kaen 40002, Thailand; kholom@kku.ac.th
4   Plant Breeding Research Center for Sustainable Agriculture, Khon Kaen University, Khon Kaen 40002, Thailand; kamol9@gmail.com
5   Department of Entomology and Plant Pathology, Faculty of Agriculture, Khon Kaen University, Khon Kaen 40002, Thailand; shanfa@kku.ac.th
6   Department of Agronomy, Iowa State University, Ames, IA 50011, USA; thomasl@iastate.edu
*   Correspondence: sphala@kku.ac.th; Tel.: +66-43-202-696

**Abstract:** The study aimed to improve the small-ear waxy corn populations for prolificacy, high total carotenoid content and resistance to downy mildew. Three cycles of modified mass selection were carried out for population improvement. Forty-four genotypes derived from eight C3 populations and six check varieties were evaluated for agronomic traits and yield at Khon Kaen and screened for downy mildew resistance in the rainy season 2020 at two experimental sites in Ban Phang district of Khon Kaen province. Fifty genotypes were clustered into six major groups based on color parameter (h°) and total ear number. Two selected groups (C and E) with two populations of small-ear waxy corn including Nei9008/BK-24-9-B and TY/TF-33-1-B were selected as they were resistant to downy mildew, prolific ears, and intense orange kernel color. Modified mass selection was effective for improvement of multiple traits in waxy corn.

**Keywords:** breeding; lutein; zeaxanthin; prolific corn; disease assessment

## 1. Introduction

Waxy corn is a popular vegetable crop in many Asian countries such as China, Korea, Indonesia, Vietnam, and Thailand [1]. It is different from normal maize due to high glutinous starch in the endosperm [2]. Among different types of waxy corn, small-ear waxy corn has many advantages including prolificacy, early maturity [3] and high carotenoid content [4]. The varieties of small-ear waxy corn currently cultivated are landraces or open-pollinated varieties (OPVs) with diverse plant types, ear shapes, and kernel colors. However, most cultivated varieties are also highly susceptible to downy mildew (DM).

DM is a major maize disease from seedling to harvest stages, resulting in an economic yield reduction up to 100% in Asia, Europe, West Africa, and North America [5,6]. Moreover, DM is more prevalent in the tropics and either a single strain or multiple strains may infect the maize plants [7]. More recently, two species of DM including *Peronosclerospora sorghi* and *P. maydis* have been reported to infect corn in the country [8]. Polygenic inheritance of DM resistance has been reported [9] and additive and non-additive gene effects were equally important in the inheritance of DM resistance in maize [10,11]. Therefore, improvement of small-ear waxy corn genotypes for DM resistance is possible.

In Thailand, small-ear waxy corn has diverse kernel pigmentations from light yellow to intense orange, and carotenoid levels from 0.89 µg g$^{-1}$ FW to 18.11 µg g$^{-1}$ FW [12]. Carotenoid-rich vegetable crops, represented by yellow to orange pigmentations, offer health benefits such as to reduce the risk of cardiovascular disease, night blindness, and cancer [13]. Provitamin A-biofortified maize, initiated by CIMMYT and the HarvestPlus project, is a successful breeding program aimed at promoting carotenoid intake and to combat vitamin A deficiency in Africa [14,15]. Currently, the color parameter (h°) can be applied through conventional breeding programs as an affordable selection tool for increasing lutein, zeaxanthin, β-carotene, and total carotenoids in waxy corn [16] and small-ear waxy corn [12].

Prolificacy is another trait of interest in small-ear waxy corn. Genotypes with two or more marketable ears per plant with a short gap period to emerge between the first and the second ears could be declared as prolific corn [3]. The recessive genetic condition mainly determines the magnitude of prolificacy [17], and some alterations could be attributed to the environment [18]. However, different studies have found different modes of inheritance which have shown that prolificacy is heritable, as indicated by the predominance of additive gene effect and high broad-sense heritability on prolificacy [19–21]. Therefore, increasing at least two marketable ears of prolific ears and the level of carotenoids, especially lutein, zeaxanthin, β-carotene, and total carotenoids, will be useful in breeding programs aiming to increase carotenoid levels and prolificacy in small-ear waxy corn.

As a classical breeding method, mass selection is popularly utilized for population improvement in maize for several quantitative traits such as downy mildew resistance [22,23], lutein and zeaxanthin contents [24,25], ear numbers per plant, and grain yield [26]. This method is simple and cost-effective. However, it can only be effective for improving the traits with high heritability and simple inheritance [27]. To address some limitations of mass selection, further improvement has been made and modified mass selection has been proposed. Modified mass selection is plant-based selection, allowing breeders to control the pollination only for selected plants within a population [28]. As a result, this method may increase the frequencies of favorable alleles and maintain genetic variability [29,30]. Previous studies have reported the positive responses to modified mass selection for prolificacy [3], ear length [31], yield [32], and plant height [33] in different corn types.

Among diverse small-ear waxy corn accessions, we noticed several genotypes with intense orange kernels, a high carotenoid content, and prolific ears [12]. However, information on DM resistance levels of selected genotypes is not available. Therefore, this study aimed to improve orange small-ear waxy corn populations by using modified mass selection for DM resistance, carotenoid content, and prolific ears. This information will be useful in the area of an applied waxy corn breeding program focused on multiple aspects such as DM resistance, carotenoid content, and ear prolificacy.

## 2. Materials and Methods

### 2.1. Plant Materials

Two field corn inbred lines and two small-ear waxy corn populations (Table 1) differing in ear prolificacy, kernel color, and DM resistance levels were used for establishing eight base populations of intense orange small-ear waxy corn. Takpha1 (Nei452008 or TF) and Nakhon Sawan1 (Nei9008) were field corn inbred lines developed by Nakhon Sawan Field Crops Research Center, Thailand [34]. These inbred lines were resistant to DM, and they have orange kernels with an intense total carotenoid content. Tein Leang Khon Kaen (TY) was an open-pollinated population developed by Plant Breeding Research Center for Sustainable Agriculture of Khon Kaen University, Thailand, while Tein Ban Kao (BK) was an open-pollinated population developed by Rajamangala University of Technology Suvarnabhumi, Thailand. TY and BK were prolific populations with small ears, yellow kernels and good eating quality at a fresh stage. However, they are highly susceptible to downy mildew.

**Table 1.** Pedigree, type of varieties, color kernel, DM resistance, and sources of genetic materials used in this study.

| Commercial Name | Code Name | Pedigree | Cultivar Type | Corn Type | Color Kernel | DMR | Source |
|---|---|---|---|---|---|---|---|
| Takpha1 | Nei452008 or TF | Pio.3003-3-2-B-3-1-4-BBB | Inbred line | Field corn | Orange | R | NSFCRC |
| Nakhon Sawan1 | Nei9008 | SW(MMS) C2F2 × Pop.28 | Inbred line | Field corn | Intense orange | R | NSFCRC |
| Tein Leang Khon Kaen | TY | Improved from Tein Leang Nhong Bae | OPV | Small-ear waxy corn | yellow | S | PBRCSA |
| Tein Ban Kao | BK | - | OPV | Small-ear waxy corn | yellow | S | RUTS |

OPV—Open pollinated variety; DMR—Downy mildew resistance; R—Resistance; S—Susceptible; NSFCRC—Nakhon Sawan Field Crops Research Center, Nakhon Sawan, Thailand; PBRCSA—Plant Breeding Research Center for Sustainable Agriculture, Khon Kaen, Thailand; RUTS—Rajamangala University of Technology Suvarnabhumi, Ayutthaya, Thailand.

### 2.2. Base Population and Population Improvement

The inbred lines were crossed to TY and BK in 2018 to obtain eight progenies (four crosses and four reciprocals). These progenies of each cross were recombined to form a base population by random mating for two consecutive seasons without selection in 2018 and 2019. Therefore, eight populations were established and further population improvement with modified mass selection was performed.

Modified mass selection was carried out for three cycles in the rainy season 2019, and in the dry season 2019/2020 at Agronomy Field Crop Station, Faculty of Agriculture, Khon Kaen University, Khon Kaen, Thailand (Figure 1). Major selection criteria through each cycle included individual plants with high prolific ears (≥2 ears per plant) and intense yellow-to-orange kernel pigmentation of the first emerged ear. As initial steps, the base populations (C0) were planted and the plants with excellent plant standing, short to moderate days to anthesis, stay green, and free-disease appearance were selected and manually self-pollinated to generate the C1 population. The selected plants were harvested and further selection of harvested dried ears was done for good husk cover and intense orange waxy kernels. Waxy kernels (*wxwx*) were identified by potassium iodide (KI) staining. The crown of dried kernels was shaved off with a lancet and was stained with an aqueous solution of potassium iodide. A homozygous waxy (*wxwx*) endosperm kernel stains reddish-brown, whereas a non-waxy endosperm kernel stains deep blue [35]. These C1 seeds among selected ears were bulked and planted in the next cycle. These procedures of selection were repeated for the next two cycles (C2 and C3 populations) and were finished in the dry season 2019/2020. Forty-four S3 lines of eight C3 populations were further evaluated for yield, yield components, agronomic traits, and downy mildew resistance in the rainy season 2020. The numbers of the S3 lines in these populations were not the same.

### 2.3. Yield Trial and Agronomic Evaluation

Four source parents (Nei452008, Nei2008, TY, and BK), forty-four S3 lines of eight C3 populations, and two OPV cultivars of small-ear waxy corn (Tein Lai-52 and Tein ayuttaya60) were assigned in a randomized complete block design (RCBD) with two replications under field conditions. The experiment was carried out in the rainy season 2020 at Agronomy Field Crop Station, Faculty of Agriculture, Khon Kaen University, Khon Kaen, Thailand (16°28′27.7″ N, 102°48′36.5″ E; 190 m above sea level). There were a total of 50 entries. The seeds were sown in two-row plots, 5 m long and with a spacing of 0.8 m between rows and 0.25 m between plants within rows.

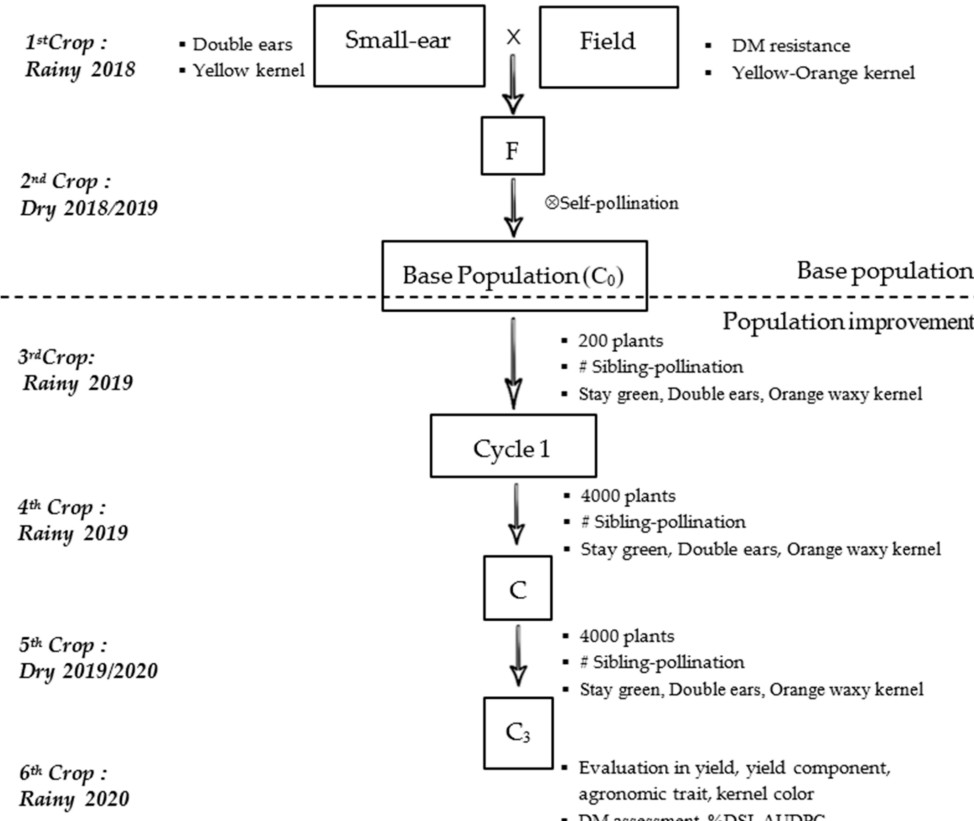

**Figure 1.** Schematic diagram for base population and population improvement of orange small-ear waxy corn populations with three cycles of modified mass selection for stay green, prolific ears, and orange waxy kernels from 2018 to 2020.

The soil was of a sandy texture and a low pH (~6.53), and contained low organic matter (0.54%) and low available soil macronutrients, especially total nitrogen (0.027%), available phosphorus (48.67 ppm), and exchangeable potassium (42.33 ppm). Accumulated rainfall during the growing season was 398.4 mm. Minimum air temperature was 21.1 °C, maximum air temperature was 40.8 °C, and relative humidity was 79.8%. The highest solar radiations during the summer seasons in 2019 and 2020 were 23 and 23.5 MJ m$^{-2}$ day$^{-1}$, respectively, which were higher than in the rainy season 2018 and in the winter 2020.

Light intensity in the summer season was higher than in both the rainy and winter seasons. Day lengths were different between seasons. Day lengths in the rainy season and the winter season were shorter than in the summer season. However, day length in summer was longer in this study period (Figure 2).

The soil was ploughed three times, and chicken manure was incorporated into the soil at a rate of 3.2 tons ha$^{-1}$ after tillage. The seeds were sown at a rate of two seeds per hill, and the seedlings were thinned to obtain one plant per hill at two weeks after sowing (WAS). Mixed fertilizer of NPK (formula 15-15-15) was applied at the rate of 312.5 kg ha$^{-1}$ before planting, and urea (formula 46-0-0) was applied at the rate of 125 kg ha$^{-1}$ at three WAS. Fall armyworm (*Spodoptera frugiperda*) was the most damaging insect in the experiment. However, fall armyworm management was controlled by using insecticides viz 20% SC Cyantraniliprole rate 20 mL per 1 kg seed for seed treatment, 5.17% SC chlorantraniliprole, 10% SC chlorfenapyr, 1.92% EC emamectin benzoate and 12% SC Spinetoram, 3 times per week for spraying from 10 days after planting until the flowering stage. A sprinkler irrigation system was available for optimum growth and yield. Sib-pollination was done for color measurements to avoid a xenia effect. Corn ears for both yield and color measurements were harvested at the fresh stage approximately 20 days after pollination (DAP).

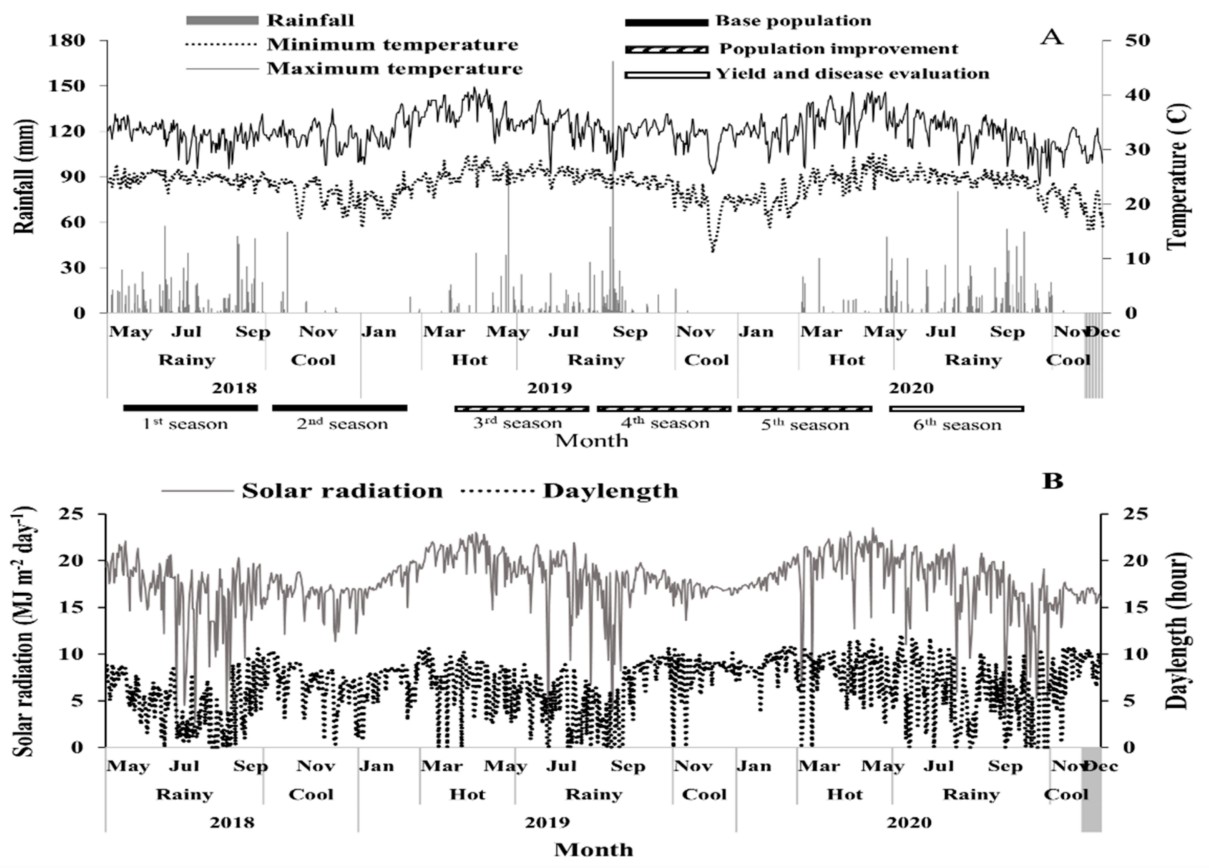

**Figure 2.** Daily rainfall, maximum and minimum temperatures (**A**), solar radiation and daylength (**B**) for population improvement and trial in the Khon–Kaen province, Thailand, 2018–2019.

Plant data were recorded for yield, yield components, and other important agronomic traits. Days to anthesis was defined as the number of days from planting to 50% of pollen shedding. Ear height was measured from ground level to the node bearing the uppermost ear after milk stage. Both traits were recorded after reproductive stage on 10 random plants per plot.

Ear yields and yield components were measured at the milk stage approximately 20 DAP. Both total husked and unhusked ear yields were partitioned to first-emerged ear yield (tons $ha^{-1}$), second-emerged ear yield (tons $ha^{-1}$), and third-emerged ear yield (tons $ha^{-1}$). Ear numbers were recorded for first, second, and third-emerged ears per plant and were converted to ear numbers per hectare. Yield components, including husked ear diameter (cm) and husked ear length (cm), were measured on ten ears per plot.

Kernel color of three ears per plot was measured with HunterLab miniscan EZ colorimeter (Mod. MSEZ-4500L, Hunter Associates Laboratory Inc., Virginia, USA). The color values were averaged from three ear points at the tip, middle, and bottom. The color parameter was indicated in hue ($h°$), with a degree unit from $0°$ to $360°$ ($0°$ = red, $90°$ = yellow, $180°$ = green, and $270°$ = blue) [36].

### 2.4. Experimental Design and Pathogen Inoculation

Nei452008 and Nei2008 are resistant to downy mildew, while TY and BK are susceptible. Tein Lai-52 and Tein ayuttaya60 were included as susceptible checks. Four parents, two checks and forty-four S3 lines from eight C3 populations were screened for resistance to downy mildew in a randomized complete block design (RCBD) with two replications under field conditions at two locations in the rainy season 2020. Two experimental sites were located in Ban Phang, Khon Kaen, Thailand with contrasting levels of downy mildew

disease severity (DMDS). At Nong Bau Village (16°27′45.2″ N 102°37′10.7″ E), DMDS was low, whereas at Sok Muang Village (16°28′14.5″ N 102°34′28.5″ E) DMDS was high.

Due to low DMDS at the first location (Nong Bau Village), a combined technique between spreader row and manual spraying was applied for pathogen inoculation and infestation [6]. Two susceptible genotypes, including Tuxpeno [37] and Niew Muang Tam (NMT), were used as spreader rows. The spreader rows consisted of one row of Tuxpeno and another row of NMT, which were planted at the border of the trial and inserted among the tested genotypes at ten-row intervals [38]. Inoculation on the spreader rows was carried out at 7 days after planting. Spore suspension of downy mildew was sprayed on the spreader rows seven times at three-day intervals. In contrast, natural inoculation can be established at the second location (Sok Muang Village) with a high DMDS. Tested lines were planted after high infection of DMD in spreader rows.

### 2.5. Disease Assessment

Disease incidence (DI), disease severity index (DSI), and area under the disease-progress curve (AUDPC) were evaluated for downy mildew resistance. Visual assessment on DMD was conducted five times (14, 21, 28, 35, and 42 DAP) at seven-day intervals. The DMD was scored using 10 rating scales: 10 = 100% of total leaves per plant having spore and mosaic symptoms; 8 = 80% of total leaves per plant showing mosaic symptoms; 6 = 60% of total leaves per plant showing mosaic symptoms with the presence of initial spore; 4 = 50% of total leaves per plant showing mosaic symptoms; 2 = 20% of leaves showing initial mosaic symptoms; 0 = no mosaic symptom [39].

The disease incidence (DI) was calculated by following formula [40] as follows:

$$DI\ (\%) = \left(\frac{\text{number of infected plants}}{\text{total number of plants}}\right) \times 100 \tag{1}$$

The disease severity index (DSI) was calculated from the disease rating by following this formula [39]:

$$DSI\ (\%) = \sum \frac{[((\text{rating no.} \times \text{no. plant in rating}) \times 100\%)]}{[(\text{total no. plant} \times \text{highest rating})]} \tag{2}$$

Next, each entry was classified as highly resistant (HR), with a 0% infection (no symptom), resistant (R) with 1–10% infection, moderately resistant (MR) with 11–25% infection, moderately susceptible (MS) with 26–50% infection, susceptible (S) with 51–75% infection, and highly susceptible (HS) with 76–100% infection [41,42].

Area under the disease-progress curve (AUDPC) was calculated as an estimate of quantitative disease resistance according to the formula described by [43]:

$$AUDPC = \sum_{i=1}^{n_{i}-1} \frac{(y_i + y_i + 1)(t_i + 1 - t_i)}{2} \tag{3}$$

where $y_i$ = leaf downy mildew severity on the i ith date; $t_i$ = ith day; and $n$ = number of dates on which disease was recorded.

### 2.6. Statistical Analysis

Data on plant response to downy mildew (DI, DSI, and AUDPC), yield, yield components, and agronomic traits were subjected to analysis of variance using STAR version 2.0.1 software [44]. Then, mean comparison among genotypes was done with the least significant difference (LSD) test using Statistix 10 [45]. The statistical model is:

$$Y_{ijk} = m + B_i + L_j + G_k + LG_{jk} + e_{ij} + e_{ijk} \tag{4}$$

where $Y_{ijk}$ was the mean of genotypes i in the location j and block k, m was mean, $B_i$ was block effects, $L_j$ was locations effects, $G_k$ was genotypes effects, $LG_{jk}$ was interaction be-

tween locations and genotypes effects, eij was location error effects, and eijk was the pooled error effects. Least significant difference (LSD) was used to compare mean differences at a 0.05 probability level. The correlation between color parameters (chroma and hue) between DI, DSI, AUDPC-DI, and AUDPC-DSI were determined by Pearson's correlation analysis. The clustering was done using the Ward criterion with the JMPPro software (version 13.0, SAS institute Inc., Chicago, IL, USA) was then performed for the color parameter (h°) and total ear number (TEN).

## 3. Results and Discussion

### 3.1. Yield and Yield Components

Genotypes were significantly different ($p \leq 0.01$ and 0.05) for all characters in this study (Table 2). Highly significant differences ($p \leq 0.01$) were observed for most characters except for husked ear length, which was significant ($p \leq 0.05$). Total unhusked ear yield had a higher variation than the first unhusked ear yield, and a similar pattern was observed for husked ear yield. For ear number, total ear had the highest variation followed by the first ear, second ear, and third ear, respectively, whereas husked ear and unhusked ear showed low variations in both ear length and ear diameter. Substantial variation was also observed for days to anthesis, plant height, and ear height. The traits with high levels of variation were interesting for the selection of superior lines.

**Table 2.** Mean squares for yield, yield components, and agronomic traits of 50 orange small-ear waxy corn genotypes in the rainy season 2020.

| Source of Variation | | Replication | Genotype | Error | CV (%) |
|---|---|---|---|---|---|
| Df | | 1 | 49 | 49 | |
| Unhusked yield | 1st | 11.6 ** | 8.3 ** | 0.2 | 8.5 |
| | total | 43.4 ** | 21.0 ** | 0.3 | 8.1 |
| Husked yield | 1st | 6.8 ** | 2.1 ** | 0.1 | 11.3 |
| | total | 17.2 ** | 5.3 ** | 0.2 | 11.7 |
| Ear number [1/] | 1st | 258.9 ** | 31.4 ** | 13.9 | 31.7 |
| | 2nd | 180.3 ** | 22.5 ** | 7.1 | 44.7 |
| | 3rd | 0.3 * | 1.1 ** | 0.1 | 66.3 |
| | total | 905.8 ** | 100.6 ** | 29.4 | 30.0 |
| Husked ear diameter | 1st | 0.3 * | 0.2 ** | 0.1 | 8.3 |
| Husked ear length | 1st | 5.6 ns | 3.3 * | 2.0 | 11.9 |
| Days to anthesis | | 185 ** | 53 ** | 21.1 | 7.9 |
| Plant height | | 9409 ** | 923 ** | 211.0 | 10.1 |
| Ear height | | 4134 ** | 497 ** | 97.0 | 13.3 |

[1/] = data in 107, 1st = the first ear position, 2nd = the second ear position, 3rd = the third ear position. *, ** significant at $p \leq 0.05$ and $p \leq 0.01$, respectively.

In previous studies, Genotype had a high contribution to total variation in unhusked ear and husked ear in small-ear waxy corn [12] and grain yield in maize [46,47]. Moreover, high levels of variation were also reported in other traits including unhusked ear weight, husked ear weight and seed yield in purple waxy corn [48] and yield in sweet corn [49]. Genotypes contributed to the largest proportion of total phenotypic variation for all traits, suggesting that total yields and ear number are selectable traits in these small-ear waxy corn populations.

### 3.2. Cluster Analysis

Based on the objective of this research, the varieties with a high carotenoid content and prolificacy are desirable. The color parameter (h°) and total ear number (TEN) were used for varieties grouping. A dendrogram based on the h° and TEN classified 50 small-ear waxy corn varieties into six major groups (A–F) (Figure 3).

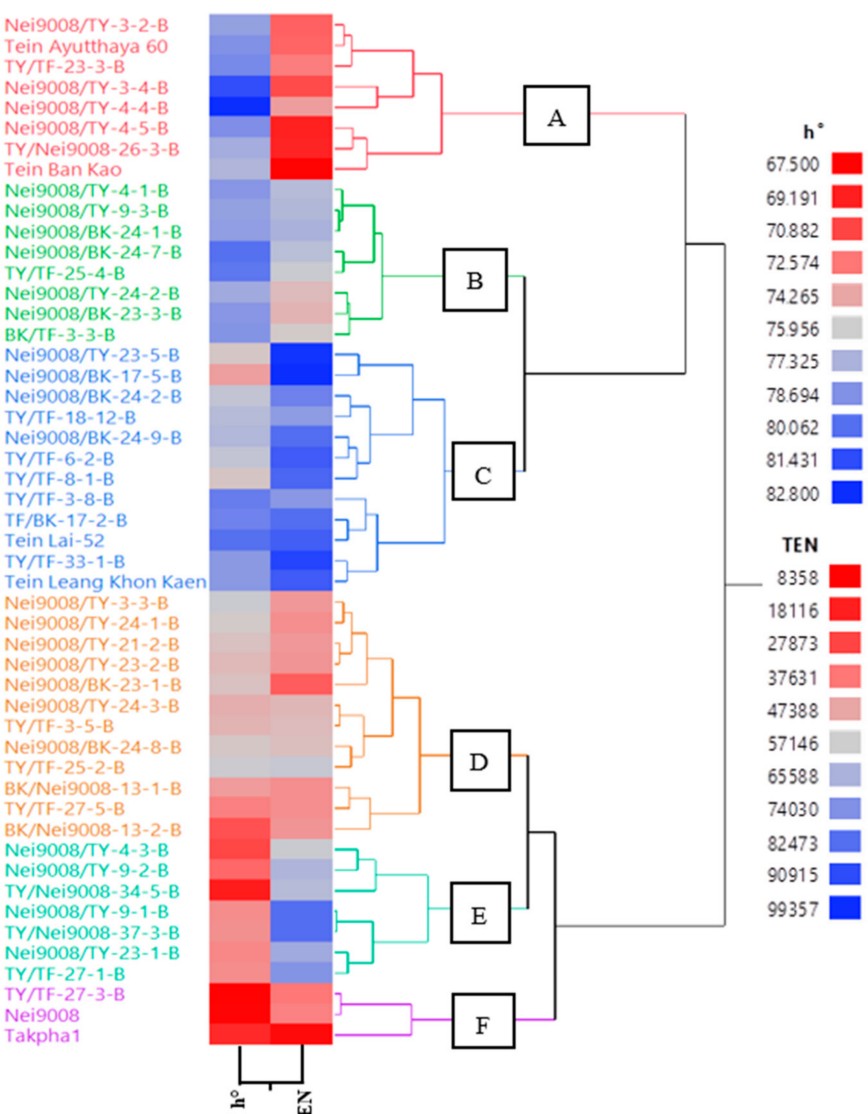

**Figure 3.** Dendrogram of phenotypic relationships among 50 small-ear waxy corn varieties. Six main clusters (**A**–**F**) were formed. Two-way Ward's cluster analysis based on color parameter (h°) and total ear number (TEN).

Group A was the small group consisting of eight varieties, namely Nei9008/TY-3-2-B, Tein Ayutthaya60, TY/TF-23-3-B, Nei9008/TY-3-4-B, Nei9008/TY-4-4-B, Nei9008/TY-4-5-B, TY/Nei9008-26-3-B and Tein Ban Kao. These varieties had moderate values of a h° (77.2–82.8) and lowest values of TEN (8358–58,500 ear ha$^{-1}$).

Group B consisted of eight varieties, namely Nei9008/TY-4-1-B, Nei9008/TY-9-3-B, Nei9008/BK-24-1-B, Nei9008/BK-24-7-B, TY/TF-25-4-B, Nei9008/TY24-2-B, Nei9008/BK23-3-B and BK/TF-3-3-B. Most maize varieties in this group had moderate to high values of a h° (77.7–80.0) and low to moderate values of TEN (50,143–65,929 ear ha$^{-1}$).

Group C was the largest and most important group, consisting of 12 varieties, namely Nei9008/TY-23-5-B, Nei9008/BK-17-5-B, Nei9008/BK-24-2-B, TY/TF-18-12-B, Nei9008/BK-24-9-B, TY/TF-6-2-B, TY/TF-8-1-B, TY/TF-8-3-8-B, TF/BK-17-2-B, Tein Lai-52, TY/TF-33-1-B, and Tein Leang Khon Kaen. However, all varieties in this group had low to moderate values of h° (74.0–80.0) and moderate to high values of TEN (71,500–99,357 ear ha$^{-1}$).

Group D contained 12 varieties, namely Nei9008/TY-3-3-B, Nei 9008/TY-24-1-B, Nei9008/TY-21-2-B, Nei9008/BK-23-1-B, Nei9008/TY-24-3-B, TY/TF-3-5-B, Nei9008/BK-24-8-B, TY/TF-25-2-B, BK/Nei9008-13-1-B, TY/TF-27-5-B and BK/Nei9008-13-2-B. All

varieties in this group showed moderate to high values of h° (71.3–76.2), and the lowest values of TEN (32,500–59,429 ear ha$^{-1}$).

Group E consisted of 7 varieties, and all varieties in this group consisted of Nei9008/TY-4-3-B, Nei9008/TY-9-2-B, TY/Nei9008-34-5-B, Nei9008/TY-9-1-B, TY/Nei9008-37-3-B, Nei 9008/TY-23-1-B and TY/TF-27-1-B showed the highest values of h° (69.1–73.5), and the highest values of TEN (58,500–82,643 ear ha$^{-1}$). This group was very important in this study.

Group F was the smallest group consisting of three varieties, namely TY/TF-27-3-B, Nei9008, and Takpha1. The varieties in this group had the highest values of h° (67.5–69.7), but they had the lowest values of TEN (12,072–39,929 ear ha$^{-1}$), and most varieties in this group except Nei9008 and Takpha1 had the highest values of h°.

Therefore, groups C and E, with the low to highest values of h° (74.0–80.0 and 69.1–73.5 respectively) and moderate to high values of TEN (71,500–99,357 and 58,500–82,643 ear ha$^{-1}$ respectively), are selected.

A previous study reported that the color parameter was closely related to total carotenoid level in orange waxy corn [16] and C0 population had 79.3 of h° linked to 9.2 µg g$^{-1}$ fresh weight for total carotenoids nearly in this study that had 77.2–82.8 of h° and C3 population had 76.5 of h° linked to 17.6 µg g$^{-1}$ fresh weight for total carotenoids [16]. Yellow small-ear waxy corn had a low level of 5.7 µg g$^{-1}$ fresh weight for total carotenoids [12]. For comparison, carrots had an extremely high level of 83.1 µg g$^{-1}$ fresh weight for total carotenoid concentration [50]. Groups C and E had prolific ears, ranging from 58,500 to 99,357 ears ha$^{-1}$ for TEN, which were similar to 88,387 ears ha$^{-1}$ for TEN of the yellow small-ear waxy corn germplasm [12]. Nevertheless, the range of ear number in this study was lower than 159,334 ears ha$^{-1}$ for TEN in a C4 population of small-ear waxy corn selected by modified mass selection [3].

Yield and agronomic traits of 15 varieties classified into six groups by cluster analysis based on color parameters and prolificacy (Figure 3) are presented in Table 3. Group C had a low to high unhusked yield and a total unhusked yield of the first ear position (4.1–12.1 and 6.0–17.2 tons ha$^{-1}$, respectively). The promising line in this group was Nei9008/BK-24-9-B with the highest unhusked yield and total unhusked yield the first ear position (12.1 and 17.2 tons ha$^{-1}$, respectively) and it was higher than six check varieties. Furthermore, most lines in this group, except TY/TF-33-1-B, had the highest husked yield and a total husked yield of the first ear position (5.9 and 8.0 tons ha$^{-1}$, respectively), which were higher than for six check varieties. However, both Nei9008/BK-24-9-B and TY/TF-33-1-B had excellent agronomic traits and good characteristics similar to those of six check varieties. Group E had an intermediate to high unhusked yield of first ear position (4.3–7.1 tons ha$^{-1}$), total unhusked yield (7.4–11.8 tons ha$^{-1}$), husked yield of first ear position (2.0–4.3 tons ha$^{-1}$), and total husked yield (3.3–5.9 tons ha$^{-1}$).

The results suggested that Nei9008/BK-24-9-B and TY/TF-33-1-B had a higher total unhusked yield than six check varieties (17.2 and 8.0 tons ha$^{-1}$, respectively). Moreover, Nei9008/BK-24-9-B had a higher total unhusked yield than 11.8 tons ha$^{-1}$ of C4 small-ear waxy corn population improved by modified mass selection, whereas TY/TF-33-1-B had a lower total unhusked yield in the previous study [3]. Although both varieties had a higher total unhusked yield than other small-ear waxy corn varieties, they were still lower than normal waxy corn (18.8 tons ha$^{-1}$) [48]. On the other hand, Nei9008/BK-24-9-B had a higher total unhusked yield than 13.3 tons ha$^{-1}$ of sweet corn (sugar 75) in Indonesia [49].

**Table 3.** Means for unhusked yield, husked yield, husked ear diameter, husked ear length days to anthesis, plant height and ear height of 17 orange small-ear waxy corn varieties and 6 check varieties in the rainy season 2020.

| Varieties | Unhusked Yield (Tons ha⁻¹) | | Husked Yield (Tons ha⁻¹) | | Ear Diameter (cm) | Ear Length (cm) | Days to Anthesis | Plant Height (cm) | Ear Height (cm) |
|---|---|---|---|---|---|---|---|---|---|
| | 1st | Total | 1st | Total | | | | | |
| Group C | | | | | | | | | |
| Nei9008/TY-23-5-B | 5.8 e–h | 10.8 c–e | 2.9 e–h | 5.3 c–e | 2.8 c–n | 13.0 a–j | 54 h–m | 182 ab | 93 bc |
| Nei9008/BK-17-5-B | 4.7 j–n | 8.9 g–i | 1.9 n–u | 3.5 i–n | 2.6 h–n | 11.6 d–k | 56 g–m | 130 h–o | 68 f–q |
| Nei9008/BK-24-2-B | 4.7 k–n | 6.0 p–s | 2.6 g–j | 3.2 k–q | 2.7 g–n | 11.8 c–k | 51 lm | 154 b–j | 73 d–p |
| TY/TF-18-12-B | 4.1 m–q | 6.8 n–p | 2.2 j–q | 3.4 i–n | 2.9 b–k | 13.5 a–h | 61 a–k | 147 c–l | 83 b–i |
| Nei9008/BK-24-9-B | 12.1 a | 17.2 a | 3.4 cd | 6.1 bc | 3.0 b–i | 14.1 a–f | 57 e–m | 157 b–h | 74 c–p |
| TY/TF-6-2-B | 5.5 f–i | 10.8 de | 1.7 q–u | 3.9 g–l | 2.5 i–o | 11.3 e–k | 56 f–m | 157 b–h | 91 b–d |
| TY/TF-8-1-B | 7.7 c | 10.9 c–e | 3.6 c | 5.1 d–f | 3.2 a–f | 14.0 a–f | 55 g–m | 201 a | 125 a |
| TY/TF-3-8-B | 5.8 e–h | 8.1 h–l | 2.8 f–i | 4.3 f–h | 2.7 f–n | 13.4 a–h | 59 b–l | 159 b–g | 90 b–e |
| TF/BK-17-2-B | 3.8 o–r | 7.0 l–p | 2.0 m–s | 3.3 j–p | 2.4 m–o | 11.1 g–k | 57 e–m | 167 b–e | 79 b–m |
| TY/TF-33-1-B | 10.4 b | 14.5 b | 5.9 a | 8.0 a | 3.1 a–g | 14.9 ab | 54 h–m | 175 a–c | 95 b |
| Group E | | | | | | | | | |
| Nei9008/TY-4-3-B | 6.3 ef | 10.6 ef | 3.3 c–e | 5.3 c–e | 2.7 h–n | 13.4 a–h | 54 h–m | 144 d–m | 61 k–s |
| Nei9008/TY-9-2-B | 7.1 cd | 9.2 gh | 4.3 b | 5.5 b–d | 3.3 a–d | 14.4 a–d | 54 h–m | 169 b–e | 80 b–k |
| TY/Nei9008-34-5-B | 4.6 k–n | 8.0 i–m | 2.0 m–s | 3.3 j–p | 2.8 f–n | 12.5 b–k | 56 g–m | 169 b–e | 94 bc |
| Nei9008/TY-9-1-B | 6.6 de | 11.8 cd | 3.0 d–g | 5.9 bc | 3.0 a–h | 13.9 a–g | 57 e–m | 159 b–h | 87 b–f |
| TY/Nei9008-37-3-B | 6.3 e–g | 9.7 fg | 3.3 c–f | 5.1 d–f | 3.0 a–h | 13.1 a–j | 54 h–m | 171 b–d | 93 bc |
| Nei9008/TY-23-1-B | 4.3 l–o | 7.4 k–o | 2.4 h–n | 4.1 g–j | 2.9 c–k | 13.5 a–h | 61 a–k | 157 b–h | 85 b–g |
| TY/TF-27-1-B | 4.7 k–n | 8.6 g–j | 2.4 h–m | 4.6 e–g | 2.9 c–k | 13.2 a–i | 54 h–m | 163 b–f | 88 b–e |
| Nei9008 | 4.3 l–o | 7.1 l–o | 2.6 g–k | 4.0 g–k | 2.4 l–o | 14.1 a–e | 67 a–d | 111 n–p | 51 q–t |
| Takpha1 | 2.8 s–u | 3.1 y–A | 0.3 x | 0.4 y | 3.3 a–e | 12.5 a–k | 67 a–c | 109 op | 56 n–t |
| Tein Leang Khan Kaen | 4.0 m–q | 6.6 n–q | 2.5 g–l | 4.1 g–j | 2.8 e–n | 11.4 e–k | 48 m | 142 d–m | 75 c–n |
| Tein Ban Kao | 1.1 x | 1.1 C | 0.4 x | 0.4 y | 2.9 b–k | 13.9 a–g | 62 a–i | 96 p | 38 t |
| Tein Lai-52 | 4.1 m–q | 6.7 n–p | 2.6 g–k | 4.1 g–j | 2.9 c–m | 11.9 c–k | 48 m | 153 b–j | 78 b–m |
| Tein Ayutthaya 60 | 2.9 s–u | 3.9 w–y | 1.5 s–v | 2.0 u–w | 2.9 b–k | 11.8 c–k | 53 i–m | 126 i–o | 63 j–s |
| CV (%) | 8.5 | 8.1 | 11.3 | 11.7 | 8.3 | 11.1 | 7.9 | 10.1 | 13.3 |

1st = the first ear position, Means followed by different letters within the same column are significantly different based on LSD at $p \leq 0.05$.

### 3.3. Disease Parameters

Combined analysis of variance for disease incidence, disease severity index, AUDPC-DI and AUDPC-DSI are showed in Table 4. The effects of location (L) were significant for most parameters, except for disease incidence (DI) at 21 and 42 days after infection. The effects of genotypes (G) were highly significant ($p = 0.01$) for most parameters, except for DI at 14 days after infection and disease severity index (DSI) at 21 days after infection.

**Table 4.** Mean squares of combined analysis of variance for disease incidence, disease severity index, AUDPC-DI and AUDPC-DSI of 5 plant ages of 23 small-ear orange waxy corn varieties in the rainy 2020 in 2 locations.

| SOV | df | 14 Days | 21 Days | 28 Days | 35 Days | 42 Days | AUDPC |
|---|---|---|---|---|---|---|---|
| | | DI (%) of the day after infected | | | | | |
| Location (L) | 1 | 212,496 ** | 65 ns | 2816 ** | 4788 ** | 411 ns | 7,275,656 ** |
| Genotype (G) | 22 | 21 ns | 622 ** | 112 ** | 407 ** | 549 ** | 55,439 ** |
| L × G | 22 | 19 ns | 191 ** | 104 ** | 238 ** | 289 ** | 45,328 ** |
| Pooled error | 44 | 23 | 88 | 49 | 33 | 13 | 11,709 |
| CV (%) | | 9.3 | 11.4 | 7.4 | 6.4 | 3.8 | 4.3 |
| | | DSI (%) of the day after infected | | | | | |
| Location (L) | 1 | 78,080 ** | 127,495 ** | 48,607 ** | 92,771 ** | 38,529 ** | 45,180,000 ** |
| Genotype (G) | 22 | 152 ** | 134 ns | 434 ** | 175 ** | 1191 ** | 362,561 ** |
| L × G | 22 | 120 ** | 138 ns | 128 ns | 168 ** | 333 ** | 105,808 ** |
| Pooled error | 44 | 50 | 120 | 83 | 40 | 71 | 29,444 |
| CV (%) | | 23.2 | 18.1 | 16.7 | 9.2 | 13.3 | 11.5 |

ns not significant. ** significant at $p \leq 0.05$ and $p \leq 0.01$, respectively. DI (%) = disease incidence, DSI (%) = disease severity index, AUDPC-DI = the area under the disease-progress curve of disease incidence, AUDPC-DSI = the area under the disease-progress curve of disease severity index 14 days.

The interactions between L and G (L × G) were significant for most parameters, except for the DI at 14 days after infection and DSI at 21 and 28 days after infection. Large differences among locations were observed for area under the disease progress curve of disease incidence (AUDPC-DI) (6,593,442 **) and area under the disease progress curve of disease severity index (AUDPC-DSI) (42,190,000 **). Genotype was also an important source of variation in days after infection of DI (112 **–622 **), AUDPC-DI (55439 **), days after infection of DSI (152 **–1191 **) and AUDPC-DSI (362,561 **). L × G interactions were an important source of variation in days after infection of DI (104 *–289 **), AUDPC-DI (45,328 **), days after infection of DSI (120 **–333 **) and AUDPC-DSI (105,808 **).

Differences among varieties were significant for most disease parameters, except for DI at 14 days after infection and DSI at 21 days after infection. The differences in these disease parameters among corn varieties are due to genetic differences. In previous studies, differences among varieties were reported for the DM infection area in sweet corn [49] and sorghum [51].

The large contribution of location to total variation for most traits was due to the influences of host, pathogen, and environment on the occurrence of a certain disease [52]. The variation among locations were generally greater than variations due to L × G interactions because of a difference in disease pressures between locations. However, significant L × G interactions suggested that genotypes performed differently between locations. The results of significant interactions were similar to those reported in resistant sorghum varieties to downy mildew disease in Uganda [51]. Similar variations among sorghum varieties in response to sorghum downy mildew have been reported [53,54].

Although L × G interactions for disease incidence, disease severity, AUDPC-DI, and AUDPC-DSI in general were smaller than those of a variety, most of them were significant, suggesting that the effects of L × G interactions could not be ignored and a test of corn varieties in many locations is still required to identify resistant varieties. The results of DI at 14 days after infection and DSI at 21 days after infection were rather confounding because of the low effects of varieties, and selection was not possible at these growth stages.

DM severity indexes of two locations (Nong Bau Village and Sok Maung Village) showing disease progress during the experiment are presented in Figure 4. At Nong Bau Village, DM severity was lower than at Sok Maung Village. The infection of downy mildew started at two weeks (during 10–17 July 2020) after planting (inoculation by manual spraying), and disease severity increased until six weeks (during 7–14 August 2020).

At Sok Maung Village, disease severity was higher than at Nong Bau Village. The disease infection was dependent solely on natural inoculation. The disease symptoms were observed as early as two weeks (during 24–31 July 2020) after planting. Disease severity index reached the highest level and decreased at four weeks (during 7–14 August 2020) after infection. However, disease severity increased again at five weeks (during 14–21 August 2020) and decreased at six weeks (during 21–28 August 2020).

The difference in disease severity between two locations would be due to the differences in environmental conditions. At Nong Bau Village, the crop was exposed to high temperature and a low air humidity during one week (7–10 July 2020) to three weeks. Therefore, low air humidity was a main cause of low infection of downy mildew.

At Sok Maung Village, the disease incidence occurred as early as one week (17–24 July 2020) after infection until the six weeks due to heavy rainfall, optimum temperature, and high relative humidity. In a previous study, the optimum conditions for infection of downy mildew included a temperature range between 10–30 °C [55], and a relative humidity higher than 80% [56].

Plant age was another factor affecting disease severity, as mature plants are more resistant than young plants. Higher disease severity at Sok Maung Village could be because of an earlier infection and a higher density of natural inoculum [57]. Similarly, the plants infected at early growth stages usually die about two weeks after infection [58].

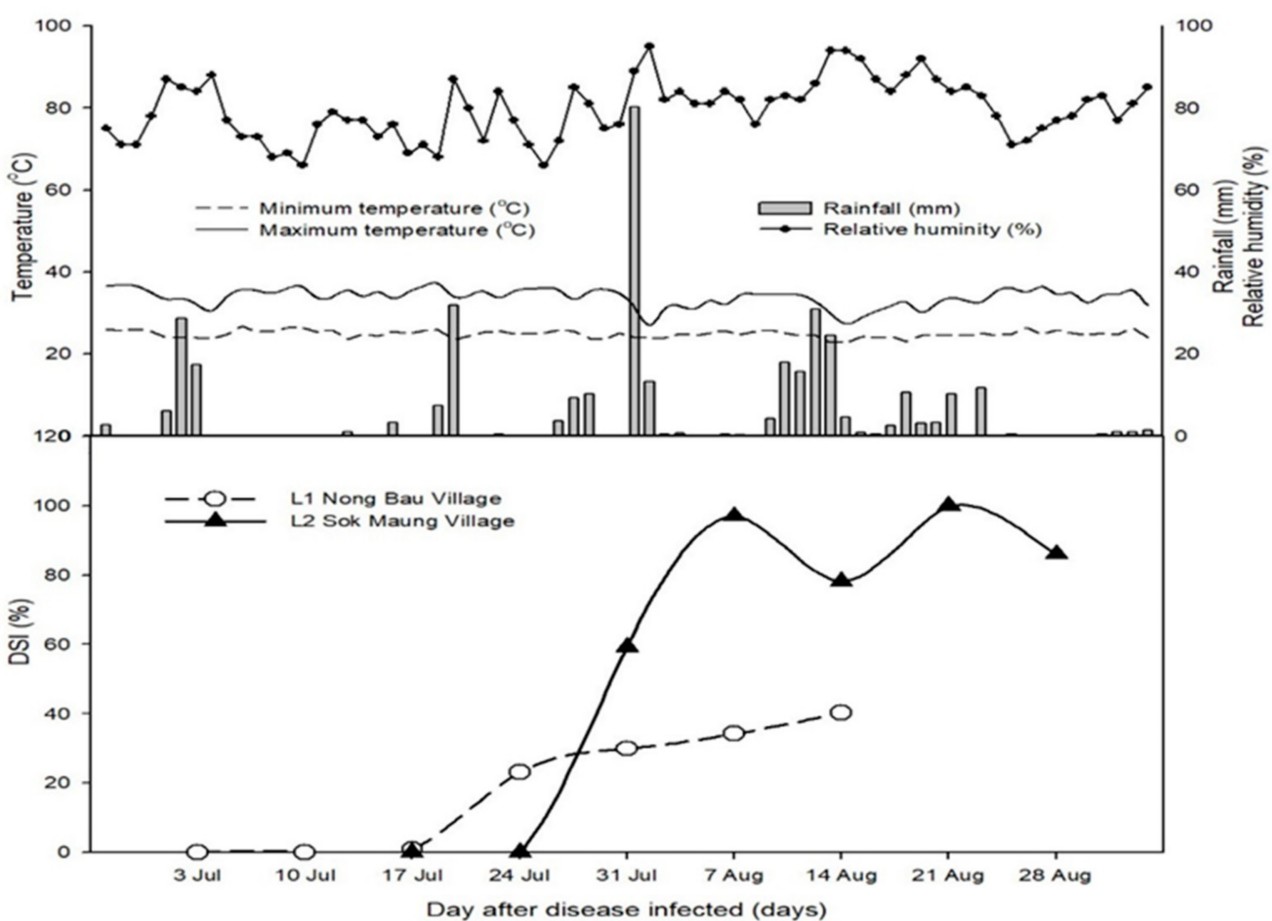

**Figure 4.** Disease severity index (%) of downy mildew in 50 small-ear waxy corn varieties at 2 locations (Nong Bau Village and Sok Maung Village), start from 3rd June 2020 until 28th August 2020, and temperature, rainfall and relative humidity at Banphang, Khon Kaen.

At Nong Bau Village, environmental conditions at early growth stages did not favor infection, and spraying of downy mildew suspension at later growth stages was necessary to increase infection levels. Inoculation at early growth stages was more effective in screening of downy mildew resistance [41]. Higher infection at Sok Maung Village would be superimposed by higher natural inoculum from waxy corn in the adjacent fields.

At Nong Bau Village, the highest DI and DSI across the twenty-one varieties were found at 42 days after infection followed by 35, 28, 21, and 14 days after infection, respectively (Supplementary Table S1). TY/TF-33-1-B had the lowest DI, DSI (Supplementary Table S1), AUDPC-DI, and AUDPC-DSI at most growth stages after infection, except DI at 35 days after infection, and it was not significantly different from check varieties (Nei9008 and Takpha1) (Figure 5). However, TY/TF-33-1-B was significantly different from four susceptible check varieties (Tein Leang Khan Kaen, Tein Ban Kao, Tein Lai-52 and Tein Ayutthaya 60) (Figure 5). Therefore, TY/TF-33-1-B showed resistance to downy mildew. Nei9008/BK-24-9-B had lower DI, DSI (Supplementary Table S1), AUDPC-DI, and AUDPC-DSI at all plant ages after infection than most varieties, except for TY/TF-33-1-B, but it was not significantly different to Takpha1 (Figure 5). On the other hand, it was significantly different to four susceptible check varieties. Therefore, Nei9008/BK-24-9-B was also identified as a moderately resistant variety (Supplementary Table S1).

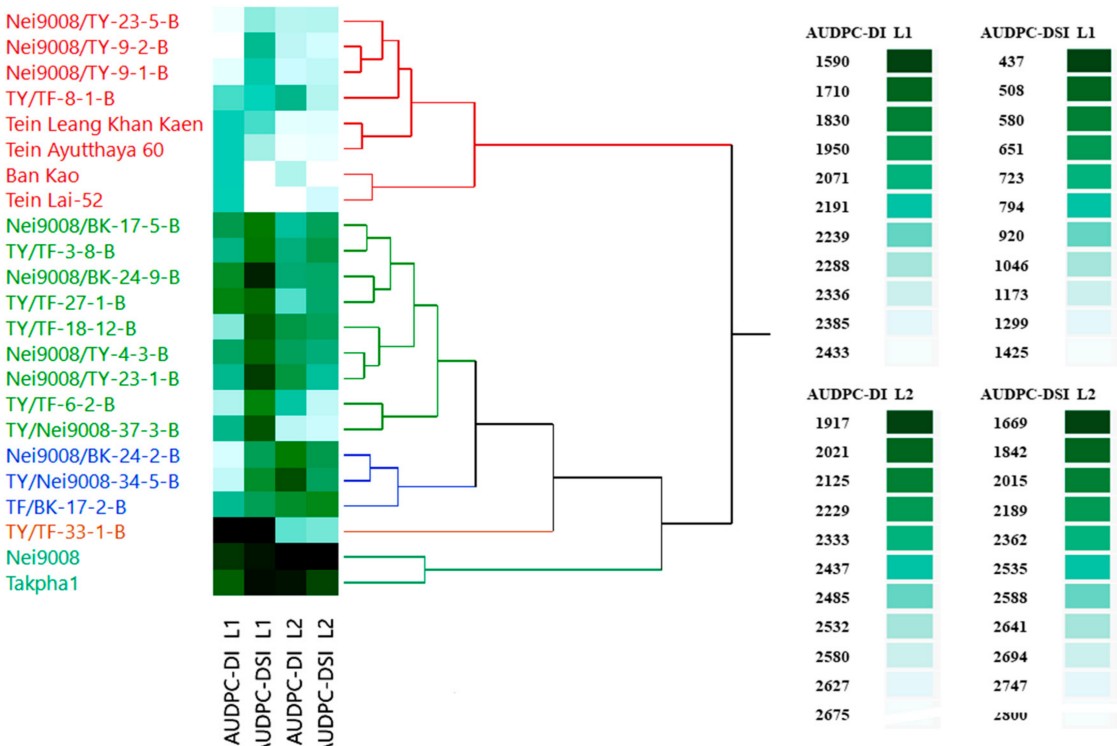

**Figure 5.** Means of the area under the disease progress curve of disease incidence (AUDPC-DI) and the disease progress curve of disease severity index (AUDPC-DSI) of downy mildew in 23 small-ear waxy corn varieties at Nong Bau Village (L1) and Sok Maung Village (L2) in the rainy season 2020.

The results in this study suggested that the crosses between resistant and susceptible varieties showed moderate resistance (Nei9008/BK-24-9-B) and resistance (TY/TF-33-1-B) to downy mildew. Difference in resistance levels among corn varieties is due to polygenic inheritance of resistance [9]. Additive and non-additive effects are equally important in the inheritance of resistance [59].

At Sok Maung Village, disease incidences at 14 and 28 days after infection were highest for most varieties (up to 100%) except Nei9008. Disease severity indexes were highest at 21 and 35 days after infection for most varieties (up to 100%) except for Nei9008 at 35 days after infection (Supplementary Table S2). DI and DSI of all varieties at Sok Maung Village were higher than those at Nong Bau Village, and AUDPC for DI and AUDPC for DSI were also higher (Figure 5).

Accordingly, DM resistance levels in all varieties at Sok Maung Village were moderately susceptible to highly susceptible. Likewise, a previous study showed that disease symptoms between early DM infections include violent stunting of infected plants with hard and steep leaves. Plants infected early usually die before the harvest stage [58].

### 3.4. Correlation between Different Plant Ages for Disease Parameters

The correlation coefficients between the data of two locations for DI at different plant ages after infection, AUDPC-DI, DSI at different plant ages after infection, and AUDPC-DSI are presented in Figure 6. DIs at all plant ages (14, 21, 28, 35, and 42 days after infection) were moderately to strongly correlated with AUDPC-DI ($r$ = 0.55, 0.77, 0.73, 0.45, and 0.67, respectively). On the other hand, most DIs at all plant ages were significantly correlated with AUDPC-DSI, except DI at 35 days after infection ($r$ = −0.18). DSIs at all plant ages were strongly correlated with AUDPC-DSI ($r$ = 0.94, 0.96, 0.95, 0.98, and 0.82, respectively). However, DSIs at all plant ages were intermediately to highly correlated with AUDPC-DI. Hence, DIs at all plant ages were moderately to strongly correlated with AUDPC-DI. In this study, the results suggested that DI and AUDPC-DI were closely associated, and these

traits can be used for the improvement of downy mildew resistance in waxy corn. However, selection efficiency should be dependent on heritability of the traits.

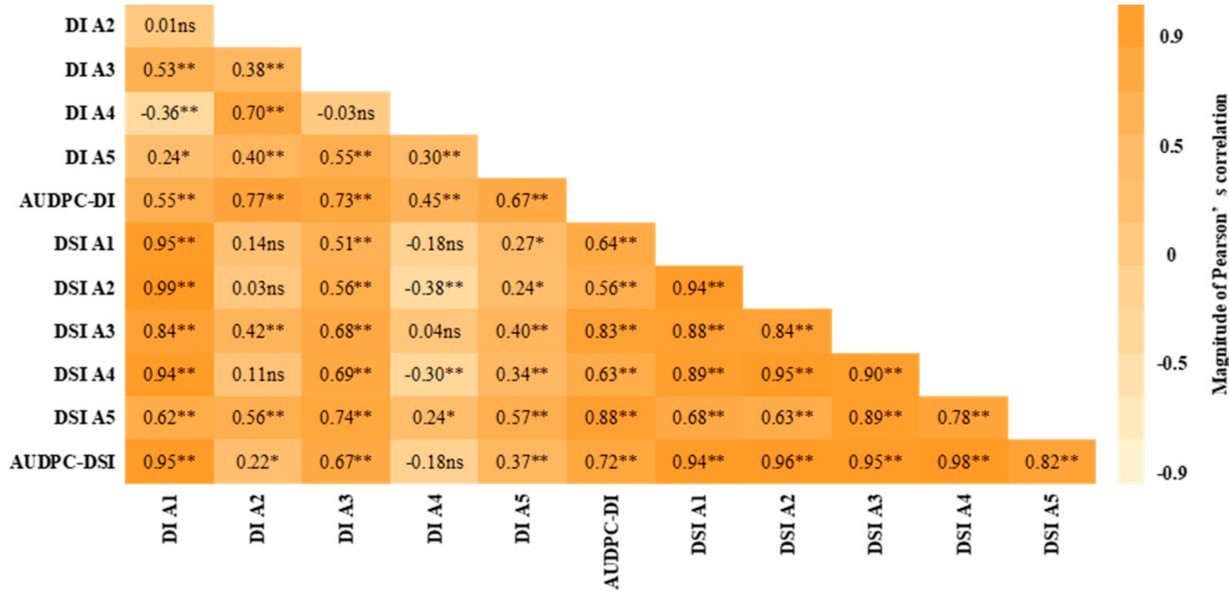

**Figure 6.** Triangular heat map representing Pearson correlation coefficients between disease incidence (DI), disease severity index (DSI), the area under the disease–progress curve of disease incidence (AUDPC-DI), and the area under the disease–progress curve of disease severity index (AUDPC-DSI) of 23 small-ear waxy corn varieties. A1= plant age 14 days after infected, A2 = plant age 21 days after infected, A3 = plant age 28 days after infected, A4 = plant age 35 days after infected, and A5 = plant age 42 days after infected. *, ** significant at $p \leq 0.05$ and $p \leq 0.01$, respectively.

DSIs at all plant ages were closely correlated with AUDPC-DSI and moderately to closely correlated with AUDPC-DI. Thus, DSI was indirectly associated with AUDPC-DSI and AUDPC-DI, and evaluation of any disease parameter was sufficient to identify disease resistance. Similar results were reported in field corn [60]. Moreover, close and positive relationships among these parameters were also reported in wheat [61] and chickpea [62].

Improvement of carotenoid content in waxy corn was the main objective of this project. Currently, it is a new trend to increase specific nutrient levels in staple food crops [63] including carotenoids in small-ear waxy corn [12]. Besides, other important traits in small-waxy corn breeding are prolificacy and resistance to downy mildew. These traits are favorable for farmers and consumers, as they contribute to higher yield and income [64]. However, better cooking and eating quality should also meet consumer preference.

Further investigations are required for the evaluation of consumer preference, such as sweetness, stickiness, tenderness, and overall liking. For human corn, more advances have been made in research on sweet corn and normal waxy corn, leaving small-ear waxy corn lags behind. In the future, small-ear waxy corn will provide not only carotenoids for daily human intake, but also a good source of income for farmers and a delicious food for consumers.

## 4. Conclusions

Forty-four S3 lines of eight C3 populations subjected to modified mass selection were classified into six major groups based on a color parameter (total carotenoid concentration) and total ear number (prolificacy). Clusters C and E had intense yellow to orange kernels and prolific ears, and both groups had two varieties (Nei9008/BK-24-9-B and TY/TF-33-1-B), with a high yield and a resistance to downy mildew. DSI was indirectly correlated with AUDPC-DSI and AUDPC-DI. Therefore, the modified mass selection can increase the

levels of total carotenoid content, prolificacy, and downy mildew resistance in small-ear waxy corn breeding programs.

**Supplementary Materials:** The following are available online at https://www.mdpi.com/article/10.3390/agronomy11091793/s1, Table S1. Means of disease incidence (DI), disease severity index (DSI), the area under the disease-progress curve of disease incidence (AUDPC-DI), the area under the disease-progress curve of disease severity index (AUDPC-DSI) and resistance levels of downy mildew of 5 plant ages in 23 small-ear waxy corn varieties at Nong Bau Village, in rainy season 2020, at Banphang, Khon Kaen. Table S2. Means for disease incidence (DI), disease severity index (DSI), the area under the disease-progress curve of disease incidence (AUDPC-DI), the area under the disease-progress curve of disease severity index (AUDPC-DSI) and resistance levels of downy mildew of 5 plant ages in 23 small-ear waxy corn varieties at Sok Maung Village, in rainy season 2020, at Banphang, Khon Kaen.

**Author Contributions:** Conceptualization, S.S., K.L. (Khomsorn Lomthaisong) and K.S.; Formal analysis, S.S., S.C., S.F. and K.S.; Methodology, S.S., J.S., S.C., K.L. (Kamol Lertrat) and K.S.; Writing—original draft, S.S., S.C., S.F., T.L. and K.S.; Writing—review and editing, S.S., J.S., S.C, T.L., K.L. (Kamol Lertrat) and K.S. All authors have read and agreed to the published version of the manuscript.

**Funding:** The Thailand Research Fund through the Royal Golden Jubilee Ph.D. Program (Grant No PHD/0214/2560).

**Data Availability Statement:** The data that support the findings of this study are available from the corresponding author upon reasonable request.

**Acknowledgments:** The research was financially supported by the Thailand Research Fund through the Royal Golden Jubilee Ph.D. Program (Grant no. PHD/0214/2560) and the Senior Research Scholar Project of Sanun Jogloy (Project no. RTA6180002). The authors would like to thank many organizations in Thailand, including the National Science and Technology Development Agency (NSTDA) (Grant no. P-20-50493), Uthai Thani Agricultural Research and Development Center, Nakhon Sawan Field Crops Research Center, Chai Nat Field Crops Research Center, Department of Agriculture (DOA), Thailand, the Plant Breeding Research Center for Sustainable Agriculture, and Rajamangala University of Technology Suvarnabhumi Huntra for providing plant materials and research facilities.

**Conflicts of Interest:** The authors declare no conflict of interest. The founding sponsors had no role in the design of the study; in the collection, analyses, or interpretation of data; in the writing of the manuscript, or in the decision to publish the results.

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
