# Peer review of "Breeding for Prolificacy, Total Carotenoids and Resistance to Downy Mildew in Small-Ear Waxy Corn by Modified Mass Selection"

_agronomy, doi:10.3390/agronomy11091793_

Round 1

Reviewer 1 Report

Line 50: …health benefits such as reduce the risk of cardiovascular disease…

Line 65: Put β-carotene as in line 55

For a better understand of this study the total carotenoid content must be included and discussed with the other authors.

Author Response

Dear Reviewer,

Thank you for your valuable comments and suggestions. We would like to submit editing manuscript in the title of Breeding for prolificacy, total carotenoids, and resistance to downy mildew in small-ear waxy corn by modified mass selection for consideration by Agronomy Journal Team. We confirm that this is the manuscript after minor revisions follow you recommend. 

Best regards,

Khundej

Reviewer 2 Report

The manuscript presented some agronomy traits of small-ear waxy corn during breeding program in Thailand. Generally, the manuscript provided valulabe information for waxy maize breeding in Asia.

Major comments:

  1. Genotyping of small-ear waxy corn: in their study, they used phenotypes to group or cluster their collection of waxy varieties/lines used. Usually, genotyping means a genetic or molecular marker-based, for example, using waxy gene. Waxy gene have been genotyped in many waxy maize population, such as in China. Only two genotypes of waxy genes were found in Chinese waxy maize varieties (Fan et al., Mol Breeding, 2008; Fan et al., PLOS ONE, 2009; Bao et al., 2012, Mol Breeding). I suggest they to do a similar work in future to real genotype their waxy maize.
  2. Too much tables (big tables) in their manuscript and they should re-organize their data and use more figures to show their data and results.

Author Response

(The authors gave the same response as above.)
